# OpenReview forum: "Agent-as-a-Judge: Evaluating Agents with Agents"
_ICLR.cc/2025/Conference — Submitted to ICLR 2025_

### Official Review · Reviewer_MHgQ · 2024-11-02

**Soundness:** 3
**Presentation:** 3
**Contribution:** 3
**Rating:** 6
**Confidence:** 4

**Summary:**

This paper introduces "Agent-as-a-Judge," a framework that uses AI agents to evaluate other AI agents' code generation capabilities, extending the existing LLM-as-a-Judge paradigm. The authors also present DevAI, a new benchmark dataset containing 55 AI development tasks with hierarchical requirements, designed to test code-generating AI systems. They evaluate three popular open-source code generation agents (MetaGPT, GPT-Pilot, and OpenDevin) using both human evaluators and their Agent-as-a-Judge system. Their results show that Agent-as-a-Judge achieves comparable performance to human evaluators (90% alignment with human consensus) while being significantly more cost-effective, reducing evaluation time by 97.72% and costs by 97.64% compared to human evaluation. The paper demonstrates that automated evaluation of AI agents is feasible and can potentially scale up the development of more sophisticated AI systems.

**Strengths:**

1. Using agent to evaluate the agent's performance is very interesting and important to advance the development of the foundation agent. This paper addresses a critical and growing challenge in AI: how to effectively evaluate increasingly complex AI agents.

2. Thorough experimental design with multiple levels of evaluation, including comprehensive ablation studies to understand component contributions, careful analysis of human evaluation biases and errors and clear comparison between human evaluators, LLM-as-a-Judge, and Agent-as-a-Judge

3. The proposed DevAI is a good benchmark, which has carefully designed hierarchical requirements, includes explicit dependencies between requirements, incorporates both mandatory requirements and optional preferences, and focuses on realistic AI development tasks

**Weaknesses:**

My main concerns of the paper are:

1. The limited technical contributions. This agent-as-a-judge is a natural extension of LLM-as-a-judge, and this is more like an engineering project.

2. The proposed DevAI is small scale with only 55 tasks, which is insufficient for a comprehensive benchmark. Though low cost is a good point, this would also over-simplify the tasks with limited task complexity and diversity.

**Questions:**

Here are some questions:

1. Could you clarify the technical contributions of this agent-as-a-judge? How this method differ from LLM-as-a-judge?

2. Could you clarify about the succeffiency of the tasks in DevAI? More tasks and evaluations should be evaluated.

3. Does this method generalize to novel tasks, which is out of the considered benchmark tasks?

---

> ### Author Response · Authors · 2024-11-20
> **Thanks for your review! Authors' feedback [1/9].**
>
> We appreciate your valuable time & insights, and thank you for highlighting the strengths of our work  (e.g., interesting and important idea, DevAI is a good benchmark, etc.). We will address your questions in the following response.
>
> ---
>
> **Q1 (original W1 (a)). This agent-as-a-judge is a natural extension of LLM-as-a-judge.**
>
> Thank you for recognizing the connection between Agent-as-a-Judge and LLM-as-a-Judge. We agree that Agent-as-a-Judge is a natural extension of LLM-as-a-Judge as we described in abstract:
> > This Agent-as-a-Judge is a natural extension of the LLM-as-a-Judge framework, incorporating agentic features that enable intermediate feedback for the entire task-solving processes for more precise evaluations.
>
> Our work identifies and addresses unique challenges inherent in evaluating agentic systems that LLM-as-a-Judge frameworks cannot adequately handle. Specifically, Agent-as-a-Judge introduces substantial technical innovations in the following areas:
>
> - **Automated Evidence Collection and Verification:** Agent-as-a-Judge autonomously collects and verifies evidence based on intermediate requirements, enabling more precise and human-aligned evaluations. For instance, it can inspect specific functions within code to ensure correct implementation, surpassing the LLM-as-a-Judge's limitation of evaluating only final outputs.
>
> - **Agentic Capabilities and Modular Design:** Designed as an agent itself, Agent-as-a-Judge possesses diverse and customizable modules, allowing it to adapt to the complex demands of evaluating agentic systems.
>
> Our goal is to conduct a methodological and thorough study of Agent-as-a-Judge, demonstrating its effectiveness in evaluating agentic systems. This work fills a critical gap left by existing LLM-as-a-Judge frameworks and provides a solid foundation for future research and applications in this area.

---

> ### Author Response · Authors · 2024-11-20
> **Thanks for your review! Authors' feedback [2/9].**
>
> **Q2 (original W2 and Q2).  The proposed DevAI is small scale with only 55 tasks, which is insufficient for a comprehensive benchmark. Though low cost is a good point, this would also over-simplify the tasks with limited task complexity and diversity. & Q2. Could you clarify about the succeffiency of the tasks in DevAI? More tasks and evaluations should be evaluated**
>
> Thank you for your insightful feedback regarding the different perspectives of the benchmark.
>
>
> **1. Scale of DevAI**
>
> Note that a concurrent benchmark for autonomous AI development, MLE-Bench [1], for example, includes 75 AI tasks with the same scale as ours.
>
> [1] Chan, Jun Shern, et al. "Mle-bench: Evaluating machine learning agents on machine learning engineering." arXiv preprint arXiv:2410.07095 (2024).
>
> We initially considered including more tasks but found that expanding beyond 55 would significantly increase evaluation costs and complexity without yielding proportional benefits in insights. For instance, OpenHands completed 55 tasks at a cost of $350.9, and GPT-Pilot took 24.78 hours to finish all tasks. Our focus was on creating a high-quality, manageable set of tasks that provide meaningful and actionable evaluations of agentic systems. By balancing cost and benefit, we believe the current number of tasks effectively distinguishes between different systems' performances and meets evaluation needs.
>
>
> | Metric                            | MetaGPT (Hong et al., 2024) | GPT-Pilot (Pythagora.io, 2023) | OpenHands (Wang et al., 2024)  |
> |------------|-------------|----------------|----------------|
> | Average Cost                      | $1.19                       | $3.92                         | $6.38                         |
> | Average Time                      | 775.29s                     | 1622.38s                        | 362.41s                       |
>
>
>
> **2. Task complexity**
>
>
> In our testing, for example, agentic systems like GPT-Pilot and OpenHands successfully resolved only 1 out of the 55 tasks by meeting all hierarchical requirements (as shown in the table below, 1.81% Task Solve Rate). This low completion rate underscores the complexity and difficulty of the benchmark, ensuring that DevAI effectively differentiates between varying levels of agent performance.
>
>
> | Metric   | MetaGPT (Hong et al., 2024) | GPT-Pilot (Pythagora.io, 2023) | OpenHands (Wang et al., 2024) |
> |---|---|---|---|
> | Requirements Met (``with dependency consideration``)  | 6.55%          |   28.96%      |        28.68%      |
> | Task Solve Rate      | 0.00%       |        1.81%    |        1.81%       |
>
>
> **3. Task Diversity**
>
>
>
> We have meticulously designed the 55 tasks in DevAI to comprehensively capture the inherent complexity and diversity of AI code generation. These tasks span a wide range of AI domains, including computer vision, natural language processing, and reinforcement learning, and encompass various model types from traditional machine learning algorithms to deep learning architectures. For example:
>
>
> - Task 15 (Image_Captioning_ShowAndTell_Flickr8k_DL): Implementing an image captioning model based on the Flickr8k dataset using the Show and Tell architecture.
>
> - Task 25 (Speech_Emotion_Recognition_CNN_LSTM_RAVDESS_DL): Develop a speech-emotion recognition system using CNN-LSTM models on the RAVDESS dataset.
>
> - Task 35  (Loan_Default_Prediction_RandomForest_LendingClub_ML): Creating a loan default prediction model using Random Forest on the LendingClub dataset.
>
>
> ---
>
> In practice, we recommend utilizing DevAI for cost-effective debugging and feedback loops when building agentic systems. For more in-depth evaluations, a full sweep of the benchmark can be conducted. In comparison, running evaluations using MLE-Bench is estimated to cost $3,000–$6,000 (a total of 1704 GPU hours, AIDE used 120.7M input tokens and 14.2M output
> tokens per task), whereas DevAI provides high-quality evaluations at a significantly lower cost (e.g., the comparatively expensive OpenHands completed 55 tasks for $350.9). This strategy effectively balances cost, complexity, and evaluation depth, offering substantial benefits over existing benchmarks.

---

> ### Author Response · Authors · 2024-11-20
> **Thanks for your review! Authors' feedback [3/9].**
>
> **Q3 (original Q1). Could you clarify the technical contributions of this agent-as-a-judge? How this method differ from LLM-as-a-judge?**
>
> Agent-as-a-Judge is specifically designed to evaluate agentic systems, while LLM-as-a-Judge is tailored for assessing language models. Our results show that this is a non-trivial extension. As discussed above (**Q1 (original W1 (a))**),  the key technical contributions (and new features) of Agent-as-a-Judge compared to LLM-as-a-Judge are: (1) Automated Evidence Collection and Verification and (2) Agentic Capabilities and Modular Design.
>
> **We give a comprehensive trajectory of Agent-as-a-Judge done in the judgment that may clearly show the specific features:**
>
> ---
>
> REQUIREMENT:
>
> ```
>    {
>         "requirement_id": 5,
>         "prerequisites": [
>             0,
>             1,
>             2,
>             3
>         ],
>         "criteria": "The regression results are visualized using \"seaborn,\" and saved to `results/figures/`.",
>         "category": "Visualization",
>   }
> ```
>
> - Automated Step 1 (**review user's query to get a comprehensive understanding**):
>
> ```
> Develop a system to predict drug response using the GDSC dataset with a Support Vector Machine (SVM) regressor. Load the dataset and perform feature selection to identify key features in `src/data_loader.py`. Implement the SVM regressor in `src/model.py`. Use cross-validation to evaluate the model's performance in `src/train.py`.  Save the performance results to `results/performance.txt`. Visualize the regression results using seaborn and save it under `results`. Next, create a report including the data preprocessing, model training, evaluation process, and the visualization. Save the report as `results/report.pdf`. The report should emphasize how feature selection impacts the model's performance, and the regression results visualization should clearly highlight the relationship between the selected features and the predicted drug response. Ensure the system is designed to be easily extendable for incorporating additional datasets or new features.
> ```
>
> - Automated Step 2 (**construct the workspace tree**):
>
> ```
>     ╭─ Project Tree ─────────────────────────────────────────────────────────────────────────────────────────────╮
>     │                                                                                                            │
>     │  Workspace Path: workspaces/OpenHands/39_Drug_Response_Prediction_SVM_GDSC_ML                              │
>     │  Total Nodes: 5                                                                                            │
>     │                                                                                                            │
>     │  Project Structure                                                                                         │
>     │  ├── .                                                                                                     │
>     │  │   └── gdsc_dataset.csv                                                                                  │
>     │  ├── results                                                                                               │
>     │  │   ├── report.md                                                                                         │
>     │  │   └── rmse_scores.png                                                                                   │
>     │  │   └── performance.txt                                                                                   │
>     │  └── src                                                                                                   │
>     │      ├── data_loader.py                                                                                    │
>     │      ├── model.py                                                                                          │
>     │      └── train.py                                                                                          │
>     │                                                                                                            │
>     ╰────────────────────────────────────────────────────────────────────────────────────────────────────────────╯
> ```

---

> ### Author Response · Authors · 2024-11-20
> **Thanks for your review! Authors' feedback [4/9].**
>
> **(Following up with Q3...)**
>
> - Automated Step 3a (**check again the relevant code snippets (a)**):
>
> ### Code Snippet
>
> **File:** `workspaces/OpenHands/39_Drug_Response_Prediction_SVM_GDSC_ML/src/train.py`
> **Lines:** `[9, 35]`
> **Identifier:** `def`
> **Category:** `function`
>
> ```python
> def evaluate_model(data_path, target_column, k=10):
>     # Load and select features
>     X, y, selected_features = load_and_select_features(data_path, target_column, k)
>
>     # Train the model
>     model = train_svm_regressor(X, y)
>
>     # Perform cross-validation
>     scores = cross_val_score(model, X, y, cv=5, scoring='neg_mean_squared_error')
>     rmse_scores = np.sqrt(-scores)
>
>     # Save performance results
>     os.makedirs('results', exist_ok=True)
>     with open('results/performance.txt', 'w') as f:
>         f.write(f"Selected features: {selected_features}\n")
>         f.write(f"Cross-validated RMSE scores: {rmse_scores}\n")
>         f.write(f"Mean RMSE: {rmse_scores.mean()}\n")
>         f.write(f"Standard deviation of RMSE: {rmse_scores.std()}\n")
>
>     # Visualize regression results
>     sns.histplot(rmse_scores, kde=True)
>     plt.title('Cross-validated RMSE scores')
>     plt.xlabel('RMSE')
>     plt.ylabel('Frequency')
>     os.makedirs('results/figures', exist_ok=True)
>     plt.savefig('results/rmse_scores.png')
>     plt.close()
> ```
>
> - Automated Step 3b (**check again the relevant code snippets (b)**):
>
> ### Code Snippet
>
> **File:** `workspaces/OpenHands/39_Drug_Response_Prediction_SVM_GDSC_ML/src/model.py`
> **Lines:** `[5, 12]`
> **Identifier:** `def`
> **Category:** `function`
>
> ```python
> def train_svm_regressor(X, y):
>     # Create a pipeline with standard scaler and SVM regressor
>     model = make_pipeline(StandardScaler(), SVR(kernel='linear'))
>
>     # Train the model
>     model.fit(X, y)
>
>     return model
> ```
>
> **...**
>
>
>
> - Automated Step 4 (**identify the related multimodal file and read it**):
>
> ```
> Content of workspaces/OpenHands/39_Drug_Response_Prediction_SVM_GDSC_ML/results/rmse_scores.png:
>
> The image is a histogram with a line plot overlayed, titled "Cross-validated RMSE scores."
>
> - **X-axis (horizontal):** Labeled "RMSE," with tick marks at 0.30, 0.32, 0.34, 0.36, and 0.38.
> - **Y-axis (vertical):** Labeled "Frequency," with tick marks from 0.00 to 2.00 in increments of 0.25.
> - **Bars:** There are three bars in the histogram.
> - The first bar is centered at approximately 0.30 with a height of 2.00.
> - The second bar is centered at approximately 0.34 with a height of 2.00.
> - The third bar is centered at approximately 0.36 with a height of 1.00.
> - **Line Plot:** A smooth blue line overlays the histogram, indicating a trend. It starts at the first bar and rises slightly over the second bar before gently declining over the third bar.
> - **Color:** The bars are shaded in a translucent blue with a black outline.
>
> The chart visually represents the distribution and trend of RMSE scores across different ranges.'
> ```

---

> ### Author Response · Authors · 2024-11-20
> **Thanks for your review! Authors' feedback [5/9].**
>
> **(Following up with Q3...)**
>
> - Automated Step 5 (**check the historical judgments (memory) on the same workspace**):
>
> ```
> --------------------------------------------------
> Requirement 1:
> --------------------------------------------------
> Criteria   : The "GDSC" drug response dataset is loaded in `src/data_loader.py`.
> Satisfied  : Yes
> Reason     :
>       <SATISFIED>: The requirement to load the "GDSC" drug response dataset in `src/data_loader.py` is satisfied. The code in `data_loader.py` includes a function `load_and_select_features` that uses `pandas` to read the dataset from a specified path (`data_path`). This function is designed to load the dataset, separate features and target, and perform feature selection, as evidenced by the code snippet provided. The function is called with a placeholder path (`'gdsc_dataset.csv'`), indicating that the dataset is intended to be loaded from this location.
> --------------------------------------------------
> ```
>
> ```
> --------------------------------------------------
> Requirement 2:
> --------------------------------------------------
> Criteria   : Feature selection is performed to identify important features in `src/data_loader.py`.
> Satisfied  : Yes
> Reason     :
>       <SATISFIED>: The requirement for performing feature selection to identify important features in `src/data_loader.py` is satisfied. The code in `data_loader.py` includes a function `load_and_select_features` that utilizes `SelectKBest` with `f_regression` to perform feature selection. This function selects the top `k` features based on their scores, as evidenced by the code snippet provided. The selected features are then returned and printed, confirming that feature selection is implemented as required.
> --------------------------------------------------
> ```
>
> ```
> --------------------------------------------------
> Requirement 3:
> --------------------------------------------------
> Criteria   : The "SVM regressor" is implemented in `src/model.py`.
> Satisfied  : Yes
> Reason     :
>       <SATISFIED>: The requirement for implementing the "SVM regressor" in `src/model.py` is satisfied. The code in `model.py` includes a function `train_svm_regressor` that creates a pipeline using `make_pipeline` with `StandardScaler` and `SVR` with a linear kernel. This setup is appropriate for training an SVM regressor, as evidenced by the code snippet provided. The function trains the model using the provided features and target values, fulfilling the requirement.
> --------------------------------------------------
> ```
>
> ```
> --------------------------------------------------
> Requirement 4:
> --------------------------------------------------
> Criteria   : Cross-validation is used to evaluate the model in `src/train.py`.
> Satisfied  : Yes
> Reason     :
>       <SATISFIED>: The requirement for using cross-validation to evaluate the model in `src/train.py` is satisfied. The code in `train.py` includes the use of `cross_val_score` from `sklearn.model_selection` to perform cross-validation with 5 folds, as indicated by the line `scores = cross_val_score(model, X, y, cv=5, scoring='neg_mean_squared_error')`. This function computes the cross-validated RMSE scores, which are then saved to `results/performance.txt`. The successful execution and saving of performance metrics, as noted in the trajectory evidence, further confirm that cross-validation is implemented and functioning as required.
> --------------------------------------------------
> ```
>
> ```
> --------------------------------------------------
> Requirement 5:
> --------------------------------------------------
> Criteria   : The performance results are saved as `results/performance.txt`.
> Satisfied  : Yes
> Reason     :
>       <SATISFIED>: The requirement to save the performance results as `results/performance.txt` is satisfied. The code in `src/train.py` includes a section where the cross-validated RMSE scores, mean RMSE, and standard deviation of RMSE are written to `results/performance.txt`, as evidenced by lines 22-26 in the code snippet. The content of `performance.txt` confirms that these metrics have been successfully saved, as it contains the selected features and the cross-validated RMSE scores, along with their mean and standard deviation.
> --------------------------------------------------
> ```

---

> ### Author Response · Authors · 2024-11-20
> **Thanks for your review! Authors' feedback [6/9].**
>
> **(Following up with Q3...)**
>
> - Automated Step 6 (**get information from the execution logs or trajectories**):
>
> ```
> ╭─ Relevant Steps in Trajectory ───────────────────────────────────────────────╮
> │                                                                              │
> │  The following environment feedback is provided for reference only and does  │
> │  not serve as decisive evidence.                                             │
> │                                                                              │
> │  - **<RELEVANT STEPS>**:                                                     │
> │                                                                              │
> │    - **Step 19**: The visualization file `rmse_scores.png` was successfully  │
> │  generated and saved in the `results/` directory. This indicates             │
> │  that the regression results were visualized using the specified tools and   │
> │  saved correctly.                                                            │
> │                                                                              │
> │    - **Step 20**: The visualization of the RMSE scores was successfully      │
> │  displayed, confirming that the visualization process using seaborn was      │
> │  executed without errors.                                                    │
> │                                                                              │
> │    - **Step 25**: A warning was encountered during the conversion of the     │
> │  Markdown report to PDF, indicating that the image                           │
> │  `rmse_scores.png` could not be fetched. This suggests a potential           │
> │  issue with the image path during the report generation process, which       │
> │  might affect the inclusion of the visualization in the final report.        │
> │                                                                              │
> │    - **Step 31**: After updating the image path to an absolute path, the     │
> │  PDF report was successfully generated, suggesting that the visualization    │
> │  was correctly referenced and included in the report.                        │
> │                                                                              │
> ╰──────────────────────────────────────────────────────────────────────────────╯
> ```
>
> After these automated steps, Agent-as-a-Judge finishes the judgment:
>
> **JUDGEMENT:** SATISFIED ✅
>
> **REASON:**
>
> The requirement to visualize the regression results using "seaborn" and save them to `results/` is satisfied. The code in `src/train.py` includes the use of `sns.histplot` from the seaborn library to create a histogram of the RMSE scores, as shown in the line:
>
> ```python
> sns.histplot(rmse_scores, kde=True)
> ```
>
> Additionally, the visualization is saved to the specified directory with the following lines:
>
> ```python
> os.makedirs('results', exist_ok=True)
> plt.savefig('results/rmse_scores.png')
> ```
>
> The presence of the file `rmse_scores.png` in the `results/` directory further confirms that the visualization was successfully created and saved, fulfilling the requirement.

---

> ### Author Response · Authors · 2024-11-20
> **Thanks for your review! Authors' feedback [7/9].**
>
> **Q4 (original W1 (b)).  This is more like an engineering project.**
>
> We agree that our work involves significant engineering efforts and a substantial manual workload, including designing the framework, carefully crafting datasets, and conducting human evaluations. As you and other reviewers have noted, this paper also identifies critical gaps and issues in evaluating agentic systems, demonstrates that existing approaches are insufficient to address these gaps, proposes a new benchmark, and presents a framework for evaluating agentic systems using agentic systems. Our work aligns with the standards of ICLR and is comparable to prior work presented at ICLR in this regard.
>
>
> **Q5 (original Q3).  Does this method generalize to novel tasks, which is out of the considered benchmark tasks?**
>
> Thank you for this important question. To assess the generalization ability of Agent-as-a-Judge, we conducted additional experiments:
>
> **1. Providing Real-Time Feedback to OpenHand:**
>
> We integrated AAAJ into OpenHands to provide intermediate feedback (we do not provide the original requirements to keep fairness), enhancing its performance on DevAI tasks. AAAJ functioned differently here—it acted as an environment feedback mechanism for the developer agent rather than merely assessing final outputs. The experimental results showed significant improvements in task completion rates with the integration of AAAJ. For example:
>
> |                     | Task 15 (Image_Captioning_ShowAndTell_Flickr8k_DL)     | Task 25 (Speech_Emotion_Recognition_CNN_LSTM_RAVDESS_DL)       | Task 35  (Loan_Default_Prediction_RandomForest_LendingClub_ML)        |
> |---------------------|--------------|-----------------|------------------|
> | **OpenHands**       | 2/6          | 3/7             | 0/7              |
> | **OpenHands + AAAJ**| 4/6 (+33.33) | 5/7 (+28.5%)    | 4/7 (+57.14%)    |
>
>
> **2. Kaggle Assistant  (we select one similar task from MLE-Bench [1]: `Facebook Recruiting III - Keyword Extraction`):**
>
> We selected a challenging task, *Facebook Recruiting III - Keyword Extraction* from MLE-Bench [1]. Without AAAJ, OpenHands could not produce any submission. With AAAJ's feedback, OpenHands successfully wrote and executed all necessary code, achieving a score comparable to a bronze medal (~0.720). This demonstrates AAAJ's ability to generalize to novel and complex tasks.

---

> ### Author Response · Authors · 2024-11-20
> **Thanks for your review! Authors' feedback [8/9].**
>
> **(Following up with Q5...)**
>
> **3.  Handling Arbitrary Queries on a Python Codebase (Codebase Grokking):**
>
> This setting uses a real, popular, and comparatively complex codebase for Agent-as-a-Judge to handle any queries about it.
>
> **Question**: What do you think about this project, and how can it be improved with real swarm intelligence? Please provide at least 5 suggestions with specific code examples where I need to add functions. Note that you do not need to consider the current `/examples` folder; please focus on the code in the `/swarm` directory. Thank you!
>
> **Project**: https://github.com/openai/swarm
>
> **Response:**
>
> The project appears to be structured around a core module named `swarm`, which likely serves as the main engine for the application. The `swarm` directory contains several key files: `util.py`, `core.py`, `types.py`, and `repl/repl.py`. To enhance this project with real swarm intelligence, here are five suggestions with specific code examples:
>
> 1. **Implement a Swarm Intelligence Algorithm**
>    - **File to Modify:** `swarm/core.py`
>    - **Suggestion:** Introduce a swarm intelligence algorithm such as Particle Swarm Optimization (PSO) or Ant Colony Optimization (ACO) to improve decision-making processes.
>    - **Code Example:**
>      ```python
>      def particle_swarm_optimization():
>          # Initialize particles
>          particles = initialize_particles()
>          for iteration in range(max_iterations):
>              for particle in particles:
>                  # Update particle velocity and position
>                  update_velocity(particle)
>                  update_position(particle)
>                  # Evaluate fitness
>                  fitness = evaluate_fitness(particle)
>                  # Update personal and global bests
>                  update_bests(particle, fitness)
>          return get_global_best(particles)
>      ```
>
> 2. **Enhance Communication Between Agents**
>    - **File to Modify:** `swarm/types.py`
>    - **Suggestion:** Define a communication protocol or message-passing interface to allow agents to share information effectively.
>    - **Code Example:**
>      ```python
>      class Message:
>          def __init__(self, sender_id, receiver_id, content):
>              self.sender_id = sender_id
>              self.receiver_id = receiver_id
>              self.content = content
>
>      def send_message(sender, receiver, content):
>          message = Message(sender.id, receiver.id, content)
>          receiver.receive_message(message)
>      ```
>
> 3. **Dynamic Task Allocation**
>    - **File to Modify:** `swarm/util.py`
>    - **Suggestion:** Implement a dynamic task allocation system where tasks are assigned based on agent capabilities and current workload.
>    - **Code Example:**
>      ```python
>      def allocate_tasks(agents, tasks):
>          for task in tasks:
>              best_agent = min(agents, key=lambda agent: agent.current_load)
>              best_agent.assign_task(task)
>      ```

---

> ### Author Response · Authors · 2024-11-20
> **Thanks for your review! Authors' feedback [9/9].**
>
> **(Following up with Q5...)**
>
> 4. **Agent Learning and Adaptation**
>    - **File to Modify:** `swarm/core.py`
>    - **Suggestion:** Introduce a learning mechanism for agents to adapt based on past experiences, using techniques like reinforcement learning.
>    - **Code Example:**
>      ```python
>      def learn_from_experience(agent, experience):
>          # Update agent's knowledge base
>          agent.knowledge_base.update(experience)
>          # Adjust strategies based on learning
>          agent.adjust_strategy()
>      ```
>
> 5. **Visualization of Swarm Activity**
>    - **File to Modify:** `swarm/repl/repl.py`
>    - **Suggestion:** Add a visualization tool to monitor and analyze the swarm's activity and performance in real-time.
>    - **Code Example:**
>      ```python
>      def visualize_swarm_activity(swarm_data):
>          import matplotlib.pyplot as plt
>          # Plot swarm data
>          plt.plot(swarm_data['time'], swarm_data['performance'])
>          plt.xlabel('Time')
>          plt.ylabel('Performance')
>          plt.title('Swarm Activity Over Time')
>          plt.show()
>      ```
>
> These enhancements aim to leverage swarm intelligence principles to improve the system's efficiency, adaptability, and overall performance. By focusing on the core `swarm` module, these suggestions can be integrated into the existing framework to create a more robust and intelligent system.
>
>
> **These experiments indicate that Agent-as-a-Judge generalizes well to novel coding tasks with minimal modifications and has the potential to be applied in various domains.**
>
> ---
>
>
> **Once again, we sincerely appreciate your valuable time. Your insights have been instrumental in helping us improve our work. We are committed to refining our submission based on your recommendations and look forward to any further discussions.**

---

> > ### Comment · Reviewer_MHgQ · 2024-11-25
> > **Thanks for the rebuttal**
> >
> > Thanks for the author's response.
> >
> > I think my main concerns are responded by the author's response, though some of them may not be that convincing to me. For example, the scale of tasks, and the technical contributions, would be beyond the capability of the authors to address during the rebuttal time.
> >
> > My initial evaluation was positive which I think is a fair evaluation so that I would keep my score.

---

> ### Author Response · Authors · 2024-11-25
> **Thanks for your feedback**
>
> **Dear Reviewer MHgQ,**
>
> ---
>
> We greatly appreciate your time and effort in reviewing our work.
>
> In response to your comments, we have carefully addressed your concerns in **Q2 (original W2 and Q2)** and **Q3 (original Q1)**.  **Based on our prior research and designed experiments, we identified key pain points:**
>
> 1. **Agentic systems should be evaluated by agentic benchmarks rather than conversational benchmarks:**
> For example, GPT-4o can achieve 90.2% on the HumanEval benchmark (initially designed for evaluating foundation models rather than agents) in a single-round conversation, while existing agentic works require significantly more resources (e.g., 1000x cost and time) to reach similar results (93%-96%). We believe such benchmarks do not effectively reflect the unique features of agentic systems (as discussed in the Introduction). To address this, we introduce the DevAI benchmark.
>
> 2. **Intermediate feedback is critical for agentic systems (but expensive):** Agentic systems operate in a physically step-by-step problem-solving manner, meaning their evaluation must consider the process, not just the final outcome. However, obtaining such feedback is costly—ideally, it should involve user experiences or expert-level judgments. We analyzed the quality, issues, and costs associated with expert-level feedback.
>
> 3. **Agent-as-a-Judge can help:** Building on points (1) and (2), after developing a meaningful benchmark and acknowledging the importance (and expense) of intermediate feedback, we explored the idea of using agents to simulate human evaluations. This approach provides a more efficient and cost-effective solution to assess agents before releasing them to the public. Our goal is to validate this direction by addressing key challenges and demonstrating practical solutions. While further improvements (e.g., multi-agent setups or more complex prompt designs) are possible, we leave them for future work. In academia, this approach provides valuable reward signals to identify bottlenecks (e.g., pinpointing issues and missed steps in key processes) before obtaining the final outcome and pave a way for recursive self-improving. In industry, it accelerates the development process by using Agent-as-a-Judge to simulate human experts or users, aiding pre-release evaluation.
>
> ---
>
> **Thank you again for your positive review score. We truly appreciate your thoughtful feedback and support in refining our work. Let us know if further adjustments are needed!**

---

### Official Review · Reviewer_4rku · 2024-11-03

**Soundness:** 2
**Presentation:** 2
**Contribution:** 2
**Rating:** 6
**Confidence:** 3

**Summary:**

The paper "Agent-as-a-Judge: Evaluating Agents with Agents" addresses the inadequacy of traditional evaluation techniques for assessing agentic systems, which require more sophisticated, step-by-step feedback mechanisms. The authors propose the Agent-as-a-Judge framework, a novel approach that employs agentic systems to evaluate other agentic systems, integrating capabilities to provide intermediate feedback throughout the task-solving process for more precise evaluations.

The paper introduces a new benchmark, DevAI, to demonstrate and validate the proposed framework. DevAI comprises 55 realistic AI code generation tasks, complete with detailed manual annotations, making it ideal for agentic evaluators. The authors benchmark three leading open-source code-generating systems—MetaGPT, GPT-Pilot, and OpenDevin—using the proposed framework. Their experiments reveal that the Agent-as-a-Judge framework outperforms the LLM-as-a-Judge method and matches the reliability of a human evaluation baseline.

The primary contributions of this work include:
1. The release of the DevAI benchmark, which consists of 55 comprehensive AI development tasks with accompanying tags, hierarchical requirements, and preferences, designed to enhance the evaluation of agentic systems.
2. The introduction of the Agent-as-a-Judge framework, an innovative method for evaluating agentic systems using other agentic systems, providing rich, intermediate feedback for more accurate evaluations that align closely with human evaluators while significantly reducing time and cost.

**Strengths:**

**Originality:**
The paper introduces the novel benchmark DevAI, which comprises 55 realistic AI code generation tasks with comprehensive manual annotations and hierarchical solution requirements. This benchmark addresses the gap in existing evaluation methods and is a significant contribution to the AI development community. The introduction of the Agent-as-a-Judge framework is another innovative aspect, proposing a novel method for evaluating agentic systems using other agentic systems. This extends the existing LLM-as-a-Judge framework by incorporating capabilities to provide intermediate feedback, thereby enhancing evaluation precision.

**Quality:**
The quality of the research is evident in the thorough experimental setup and robust analysis. The authors have benchmarked three leading open-source code-generating systems—MetaGPT, GPT-Pilot, and OpenDevin—using the proposed framework. They conducted experiments across various settings, including black-box and gray-box scenarios, as well as independent and task-dependent evaluations. These comprehensive experiments validate the effectiveness of the Agent-as-a-Judge framework and underscore its superiority over traditional methods. The detailed statistical analysis and alignment with human evaluators further reinforce the reliability of the proposed method.

**Clarity:**
The paper is well-structured and clearly presents its ideas and contributions. The introduction effectively outlines the motivation and objectives, while subsequent sections provide a detailed explanation of the DevAI benchmark and the Agent-as-a-Judge framework. The methodologies and experiments are described with clarity, making it easy for readers to follow and understand the research. Additionally, the inclusion of figures and tables enhances the presentation by providing visual insights into the experimental results.

**Significance:**
The significance of the paper lies in its potential to transform the evaluation of agentic systems. By introducing a novel and possibly more sound method—Agent-as-a-Judge—the paper addresses the limitations of traditional evaluation techniques that focus solely on final outcomes or require excessive manual labor. The DevAI benchmark, coupled with the Agent-as-a-Judge framework, provides a comprehensive and efficient approach for assessing the performance of agentic systems. This work lays the groundwork for more precise, intermediate evaluations, which can significantly accelerate progress in the development and deployment of sophisticated agentic systems.

**Weaknesses:**

**Human Evaluation Methodology :**
The use of only three human experts as judges raises concerns about the robustness and reliability of the evaluation process. Given that these experts are authors of the paper, there is a potential for bias, and the large disagreements observed among them further question the dependability of the evaluation. To improve this aspect, the study could benefit from recruiting a larger and more diverse pool of external evaluators. Even applying a subset of tasks to a larger group of judges would offer more statistically reliable results and help validate the labels in the DevAI benchmark. This would ensure that the benchmark has a more concrete "ground truth", thus making the subsequent results more robust and credible.

**Inconsistent Statements on Human Evaluation Reliability :**
There are conflicting statements regarding the reliability of human evaluations. In Section 4 (Line 366), it is mentioned that "Human evaluation, while somewhat reliable," which seems inconsistent with the prior conclusions in Section 3.4 (Line 354) "Human judgment errors are inevitable" and the observations in Lines 319-327 about the large disagreements among human evaluators. The paper would benefit from clearly articulating the relative reliability of various evaluation methods. If I understand correctly, the paper is expressing the relationship of reliability as LLM-as-a-Judge < Single-Human-as-a-Judge < Agent-as-a-Judge < Ensemble of Human Judges. The paper would benefit from stating this clearly.

**Clarity in Figures :**
Figure 2(1) appears unprofessional and is highly unreadable due to the mixed vertical and horizontal text as well as the small font size. Enhancing the readability and professional presentation of this figure is essential.  the authors might consider using a different type of diagram to display the word frequency more effectively.

**Citation Issues :**
There are several inconsistencies and inaccuracies in the citations. For instance, the citation of SWE-Bench in the introduction leads to SWE-Agent, which is incorrect. Moreover, some papers that have been accepted to conferences are cited in their arXiv versions (e.g., SWE-Bench, DSPy, HumanEval, AgentBench). Ensuring that all citations are accurate and up-to-date will enhance the paper's credibility. Each referenced work should be properly cited in its correct and most formal publication form when available.

**Questions:**

- I'm not exactly sure of the meaning of independent tasks (I) and tasks considering task dependencies (D). Does R0 belong to (I) and R1, R2 belong to (D) in Figure 1? Maybe I missed something, could you please further clarify this?
- In line 486, how is Table 3 able to demonstrate the usefulness of the "retrieve" module?
- What is your backbone model for the LLM-as-a-Judge and Agent-as-a-Judge frameworks? Does using a different backbone affect your main results?

Please also pay attention to the questions and suggestions mentioned in the "Weaknesses" section.

---

> ### Author Response · Authors · 2024-11-20
> **Thanks for your review! Authors' feedback [1/5].**
>
> We appreciate your valuable time & insights, and thank you for highlighting the strengths of our work (e.g., novel benchmark, thorough experiments and analysis, well-structured, etc.). We  will address your questions in the following response.
>
> ---
>
> **Q1 (original W1).  To improve this aspect, the study could benefit from recruiting a larger and more diverse pool of external evaluators. Even applying a subset of tasks to a larger group of judges would offer more statistically reliable results and help validate the labels in the DevAI benchmark. This would ensure that the benchmark has a more concrete "ground truth", thus making the subsequent results more robust and credible.**
>
>
> Thank you for this insightful comment. We understand that involving a larger and more diverse group of evaluators could enhance our study. However, due to the significant time commitment required (approximately 19 hours per expert) and the necessity for expert qualifications, assembling and managing a large expert panel for a full evaluation presented challenges.
>
> However, to address your concern, we conducted an additional study with 10 MSc and PhD students in AI-related fields on 7 randomly selected tasks (about 12.7% of DevAI) to evaluate OpenHands's performance. The results are summarized below:
>
>
>
>
> | Evaluation Panel            | Majority Vote Result (Alignment with Previous Consensus Result) (%) | Majority Vote Result (Alignment with Previous Majority Vote) (%) | Average Time per Person (hrs) | Average Cost per Person ($) |
> |-----------------------------|-------------------------|---------------------------------------------------------|-------------------------------|-----------------------------|
> | Larger Panel (10 experts)   | 95.23                                                    | 97.67                                                  | 1.13                         | 15.20                      |
>
>
> In this additional evaluation, the majority vote alignment (with 10 experts) with our previous majority vote results (with 3 experts) was 97.67%. This consistency reinforces the reliability of our initial evaluation and suggests that AAAJ's performance is comparable to that of a broader human panel. We observe that compared to the previous majority voting results with three human experts (92.85% for the same 7 tasks, totaling 42 requirements), the extended study achieved a modest 2.38% improvement in alignment, which demonstrates more experts involved in the majority vote may improve the alignment rate (L358-LL365). After checking the disagreement between the majority vote results from the larger panel of experts and our previous consensus results, we found that consensus results are accurate. We suggest this is due to the natural features of human brainstorming may be more effective in correcting errors and biases.

---

> ### Author Response · Authors · 2024-11-20
> **Thanks for your review! Authors' feedback [2/5].**
>
> **Q2 (original W2 (a)). There are conflicting statements regarding the reliability of human evaluations. In Section 4 (Line 366), it is mentioned that "Human evaluation, while somewhat reliable," which seems inconsistent with the prior conclusions in Section 3.4 (Line 354) "Human judgment errors are inevitable" and the observations in Lines 319-327 about the large disagreements among human evaluators.**
>
>
> Thank you for bringing this to our attention. We understand that the statements may seem conflicting at first glance, but they are consistent when considering the scope of our evaluation. Our focus is on assessing the alignment rate between Agent-as-a-Judge and Human-as-a-Judge, with the understanding that if Agent-as-a-Judge aligns closely with Human-as-a-Judge, it can be considered sufficiently reliable.
>
> The statement ``Human evaluation, while somewhat reliable`` reflects the fact that human judgments are generally regarded as more reliable than automated methods and often serve as the ``gold standard.`` However, absolute ground truth is inherently difficult to obtain due to the biases and errors inherent in human judgment. Studies have shown significant disagreement among human annotators, with inter-annotator agreement levels typically ranging from 0.6 to 0.7 [1, 2]. This underscores the inherent limitations of human evaluation, despite its recognized value.
>
> As noted in Section 3.4, ``human judgment errors are inevitable.`` Therefore, we treat human judgments as a reliable baseline for comparison, rather than as perfect ground truths.
>
> [1] Ouyang, Long, et al. "Training language models to follow instructions with human feedback." Advances in Neural Information Processing Systems 35 (2022): 27730-27744.
>
> [2] Wang, Binghai, et al. "Secrets of RLHF in large language models Part II: Reward modeling." arXiv preprint arXiv:2401.06080 (2024).
>
>
> **Q3 (original W2 (b)). The paper would benefit from clearly articulating the relative reliability of various evaluation methods. If I understand correctly, the paper is expressing the relationship of reliability as LLM-as-a-Judge < Single-Human-as-a-Judge < Agent-as-a-Judge < Ensemble of Human Judges. The paper would benefit from stating this clearly.**
>
>
> Yes, your understanding is correct, and we appreciate the suggestion. To improve clarity, we have added the following statement to the manuscript: Our observations indicate the relative reliability of evaluation methods as:
> ``LLM-as-a-Judge < Single-Human-as-a-Judge < Agent-as-a-Judge < Ensemble of Human Judges. Future advancements in foundation models and Agent-as-a-Judge designs may shift this order.``

---

> ### Author Response · Authors · 2024-11-20
> **Thanks for your review! Authors' feedback [3/5].**
>
> **Q4 (original W3). Figure 2 (1) appears unprofessional and is highly unreadable due to the mixed vertical and horizontal text as well as the small font size. Enhancing the readability and professional presentation of this figure is essential. the authors might consider using a different type of diagram to display the word frequency more effectively.**
>
> Upon reinspection, we agree with the reviewer here. According to this suggestion, we conduct an analysis to display the word frequency (the table below). This table highlights the most frequent technical words that appear in the queries, reflecting the dataset features in AI developments. Words like *dataset*, *report*, *feature*, *results*, *model*, and *data* stand out prominently.
>
> | Word         | Frequency |
> |--------------|-----------|
> | results      | 194       |
> | model        | 161       |
> | src          | 148       |
> | save         | 129       |
> | dataset      | 85        |
> | figures      | 66        |
> | data_loader  | 62        |
> | data         | 60        |
> | report       | 55        |
> | system       | 53        |
> | feature      | 36        |
>
>
> **Q5 (original W4). Citation Issues: There are several inconsistencies and inaccuracies in the citations. For instance, the citation of SWE-Bench in the introduction leads to SWE-Agent, which is incorrect. Moreover, some papers that have been accepted to conferences are cited in their arXiv versions (e.g., SWE-Bench, DSPy, HumanEval, AgentBench). Ensuring that all citations are accurate and up-to-date will enhance the paper's credibility. Each referenced work should be properly cited in its correct and most formal publication form when available.**
>
> We have thoroughly reviewed all the citations and corrected the inconsistencies and inaccuracies (updating 19 references). We will ensure that all references are accurate and up-to-date in the final version of the paper.

---

> ### Author Response · Authors · 2024-11-20
> **Thanks for your review! Authors' feedback [4/5].**
>
> **Q6 (original Q1). I'm not exactly sure of the meaning of independent tasks (I) and tasks considering task dependencies (D). Does R0 belong to (I) and R1, R2 belong to (D) in Figure 1? Maybe I missed something, could you please further clarify this?**
>
> Certainly. In our context, we evaluate the DevAI tasks in two ways (``(I) without considering dependencies with different requirements, (D) considering dependencies with different requirements``):
>
>
> Here is an example task with the intermediate requirements:
>
> ```
> "name": "39_Drug_Response_Prediction_SVM_GDSC_ML",
>
> "query": "Develop a system to predict drug response using the GDSC dataset with a Support Vector Machine (SVM) regressor. Load the dataset and perform feature selection to identify key features in `src/data_loader.py`. Implement the SVM regressor in `src/model.py`. Use cross-validation to evaluate the model's performance in `src/train.py`.  Save the performance results to `results/metrics/performance.txt`. Visualize the regression results using seaborn and save it under `results/figures/`. Next, create a report including the data preprocessing, model training, evaluation process, and the visualization. Save the report as `results/drug_response_prediction_report.pdf`. The report should emphasize how feature selection impacts the model's performance, and the regression results visualization should clearly highlight the relationship between the selected features and the predicted drug response. Ensure the system is designed to be easily extendable for incorporating additional datasets or new features.",
>
> "requirements": [
>     {
>         "requirement_id": 0,
>         "prerequisites": [],
>         "criteria": "The \"GDSC\" drug response dataset is loaded in `src/data_loader.py`.",
>         "category": "Dataset or Environment"
>     },
>     {
>         "requirement_id": 1,
>         "prerequisites": [0],
>         "criteria": "Feature selection is performed to identify important features in `src/data_loader.py`.",
>         "category": "Data preprocessing and postprocessing"
>     },
>     {
>         "requirement_id": 2,
>         "prerequisites": [],
>         "criteria": "The \"SVM regressor\" is implemented in `src/model.py`.",
>         "category": "Machine Learning Method"
>     },
>     {
>         "requirement_id": 3,
>         "prerequisites": [1, 2],
>         "criteria": "Cross-validation is used to evaluate the model in `src/train.py`.",
>         "category": "Performance Metrics"
>     },
>     {
>         "requirement_id": 4,
>         "prerequisites": [0, 1, 2, 3],
>         "criteria": "The performance results are saved as `results/metrics/performance.txt`.",
>         "category": "Performance Metrics"
>     },
>     {
>         "requirement_id": 5,
>         "prerequisites": [0, 1, 2, 3],
>         "criteria": "The regression results are visualized using \"seaborn,\" and saved to `results/figures/`.",
>         "category": "Visualization"
>     },
>     {
>         "requirement_id": 6,
>         "prerequisites": [0, 1, 2, 3, 4, 5],
>         "criteria": "A report containing data preprocessing, model training, evaluation process, and the regression results visualization, is created and saved as `results/drug_response_prediction_report.pdf`.",
>         "category": "Other"
>     }
> ]
> ```
>
> Therefore:
>
> - ``Requirements Met (I)`` means *we evaluate the intermediate requirements/tasks while ignoring the "prerequisites" of other requirements/tasks*. This allows for, say, the implementation of the learning algorithm without having a correct implementation of a data loader.
> - ``Requirements Met (D)`` means *we evaluate the requirements/tasks with the requirements that the "prerequisites" of a requirement be met before that requirement could be considered satisfied.* For example, if requirement_2 was implemented correctly, but requirement_1 is a prerequisite of requirement_2 and is not implemented correctly, then we consider requirement_2 to not have been satisfied.

---

> ### Author Response · Authors · 2024-11-20
> **Thanks for your review! Authors' feedback [5/5].**
>
> **Q7 (original Q2). In line 486, how is Table 3 able to demonstrate the usefulness of the "retrieve" module?**
>
> Sorry, this is an unclear illustration. Unlike the black-box setting, the gray-box setting allows for the retrieve module and the impact on performance of this can be partially seen in Table 3, as the trajectory provides additional valuable information. We make this clearer in the updated manuscript: ``Adding retrieve does not always provide a significant benefit in this case. We found the retrieve module effective for judging MetaGPT and GPT-Pilot, as it provides valuable trajectory information (as shown in Table 3. However, it is less effective for OpenHands, which sometimes fails to execute files, resulting in missing responses. In such cases, judgment without trajectories remains viable.``
>
>
> **Q8 (original Q3). What is your backbone model for the LLM-as-a-Judge and Agent-as-a-Judge frameworks? Does using a different backbone affect your main results?**
>
> Excellent question.  In our initial experiments, we used ``gpt-4o-2024-0513`` as the backbone model. To assess the impact of different backbones on the performance of Agent-as-a-Judge, we conducted an ablation study. The results are below.
>
> | Model                | Version                    | Parameters | agent-as-a-judge alignments with human-as-a-judge |
> |----------------------|----------------------------|------------|---------------------------------------------------|
> | LLAMA [1]            | 3.2                        | 90B        | 87.76%                                            |
> | Qwen [2]             | Coder 2.5                  | 32B        | 88.73%                                            |
> | Claude [3]           | claude-3-5-sonnet-20241022 | Unknown    | 92.95%                                            |
> | ChatGPT (Submission) | gpt-4o-2024-0513           | Unknown    | 90.16%                                            |
>
> These results allow us to conclude that the backbone does have a meaningful effect on the alignment but a relatively marginal one. We found that Claude's results are better than what we reported in the submitted manuscript, we hypothesize this is because ``claude-3-5-sonnet-20241022`` has been trained with strong function calling skills and agentic features.
>
>
> [1] https://ai.meta.com/blog/llama-3-2-connect-2024-vision-edge-mobile-devices/
>
> [2] https://qwenlm.github.io/blog/qwen2.5-coder-family/
>
> [3] https://www.anthropic.com/claude/sonnet
>
> ---
>
> **Once again, we sincerely appreciate your valuable time. Your insights have been instrumental in helping us improve our work. We are committed to refining our submission based on your recommendations and look forward to any further discussions.**

---

> > ### Comment · Reviewer_4rku · 2024-11-21
> >
> > Thank you for your response. I will raise my score for Q1 and Q7. I kindly recommend that the authors include these two parts, or at least the main conclusions and a link to the appendix, in the main body of the paper. These results might be beneficial to other readers as well.
> >
> > Additionally, I would like to suggest that the authors further refine their paper for clarity and readability. I experienced the same confusion mentioned by Reviewer qJsZ (weakness 1) when I first read this paper. Perhaps incorporating some leading sentences at the beginning of each section would help (just a thought; you could explore other ways to enhance clarity).

---

> > > ### Author Response · Authors · 2024-11-25
> > > **Thanks again**
> > >
> > > **Dear Reviwer 4rku,**
> > >
> > > ---
> > >
> > > We greatly appreciate your time and effort in reviewing our work.
> > >
> > > In response to your suggestions, we have revised the submission to enhance its clarity and readability. Specifically, we have added leading sentences and refined the explanations to make the content more accessible.
> > >
> > > ---
> > >
> > > **Thank you for your positive review score.**

---

### Official Review · Reviewer_qJsZ · 2024-11-04

**Soundness:** 2
**Presentation:** 1
**Contribution:** 3
**Rating:** 5
**Confidence:** 3

**Summary:**

This paper introduces the Agent-as-a-Judge framework, an innovative approach for evaluating agentic systems by employing other agentic systems as evaluators. Building on prior methods like LLM-as-a-Judge, this framework provides intermediate feedback throughout the problem-solving process, moving beyond final outcome evaluation. The paper applies this approach to code generation tasks, introducing the DevAI dataset, consisting of 55 realistic development tasks with hierarchical solution requirements. The results show that Agent-as-a-Judge aligns closely with human evaluation and outperforms LLM-as-a-Judge in task accuracy and cost-effectiveness.

**Strengths:**

1. The introduction of the Agent-as-a-Judge framework addresses a critical gap in evaluating agentic systems by enabling feedback at each stage of task completion, making it especially suited for complex, multistep tasks.
2. The DevAI dataset provides a practical and comprehensive testbed for agentic systems, encompassing various AI development tasks that closely mirror real-world demands, which enhances the relevance of the evaluation.
3. The study compares Agent-as-a-Judge with both human evaluators and LLM-as-a-Judge, demonstrating superior alignment with human consensus while significantly reducing costs and evaluation time.
4. The paper includes extensive ablation studies, cost analysis, and performance metrics, providing insights into the efficacy of different components of the framework and supporting reproducibility.

**Weaknesses:**

1. The overall writing is inconsistent and confusing: in the abstract and introduction,  the Agent-as-a-Judge (AAAJ) framework is proposed with its novelty and effectiveness being emphasized. But in Section 2 and 3, the paper suddenly turns to the DevAI benchmark, introducing the motivation and technical details of this particular benchmark. The AAAJ benchmark is not comprehensively introduced until Section 4.  I feel like these two contributions, namely the AAAJ framework and the DevAI benchmark, should have been separated into two papers and discussed in more details respectively. Otherwise, it may be better to start from the DevAI benchmark, with agentic evaluation framework being one of its useful features.
2. Among the four principal contributions, the top two seem to be deviated from the main idea of this paper.
3. In Section 4, AAAJ is only discussed as a proof-of-concept, with little technical details.
4. Code generation is only one of the many applications of agentic systems. There are plenty of other domains where LLM agents may help. It is therefore not enough to consider the task of code generation alone. The agentic evaluation framework also seems to be ad hoc to code generation.

**Questions:**

1. While the AAAJ framework achieves higher alignment rate than other baseline algorithms, chances are that AAAJ makes incorrect evaluations. Did you conduct any studies on the failure cases to see whether there are consistent patterns that may be exploited by agentic systems?
2. Can the AAAJ framework be evaluated in benchmarks other than code generation?

---

> ### Author Response · Authors · 2024-11-20
> **Thanks for your review! Authors' feedback [1/4].**
>
> We appreciate your valuable time & insights, and thank you for highlighting the strengths of our work (e.g., address a critical gap, practical and comprehensive testbed, significantly reducing costs and evaluation time, etc.). We will address your questions in the following response.
>
> ---
>
> **Q1 (original W1 (a)). In the abstract and introduction, the Agent-as-a-Judge (AAAJ) framework is proposed with its novelty and effectiveness being emphasized. But in Section 2 and 3, the paper suddenly turns to the DevAI benchmark, introducing the motivation and technical details of this particular benchmark.**
>
> Thank you for highlighting this structural concern. Our intention was to organize the paper following the logical sequence we adopted during our research process. Namely that **(1)** no sufficient benchmarks exist (thus requiring the creation of DevAI), **(2)** that a sufficient benchmark required extensive human effort to evaluate (shown with our human evaluation), and, finally, **(3)** how an AAAJ framework solves the issue encountered in **(2)**.
>
> According to your comments and suggestions, we now add a paragraph in the introduction and emphasize how Section 2 introduces DevAI to address the lack of benchmarks, Section 3 establishes Human-as-a-Judge as a baseline highlighting evaluation challenges, and Section 4 presents AAAJ as a scalable solution: ``This paper is structured as follows: **Sec 2** introduces DevAI to address the lack of benchmarks for verifying agentic systems with intermediate processes. **Sec 3**  establishes Human-as-a-Judge as a manual evaluation baseline, highlighting its limitations. Finally, **Sec 4**  presents Agent-as-a-Judge, a scalable solution to these challenges. More details are provided in **Appendix A and B**.``

---

> ### Author Response · Authors · 2024-11-20
> **Thanks for your review! Authors' feedback [2/4].**
>
> **Q2 (original W1 (b) and W2). I feel like these two contributions, namely the AAAJ framework and the DevAI benchmark, should have been separated into two papers and discussed in more details respectively. Otherwise, it may be better to start from the DevAI benchmark, with agentic evaluation framework being one of its useful features;  Principal contributions deviation.**
>
> We understand your perspective. While these two contributions might seem to belong to separate papers, they are intrinsically linked and mutually supportive. For DevAI to be practically usable, we suggest having AAAJ (due to the cost of human evaluations). Likewise, for AAAJ to be useful, we suggest having a benchmark that allows for such complex evaluations (which has been something largely not attended to over time due to issues with the cost of evaluation).
>
> This work thus opens the way for further development of similar benchmarks to DevAI with the agentic evaluation systems attached (i.e., AAAJ).
>
>
>
> **Q3 (original W3).  In Section 4, AAAJ is only discussed as a proof-of-concept, with little technical details.**
>
>
> We appreciate the importance of providing detailed technical information. While our primary aim in Section 4 was to introduce the design paradigm of the AAAJ system and share insights from its development, we have included detailed technical information in Appendices K–M (6 pages), covering code descriptions and prompt usage. We will ensure to reference these appendices more clearly in the main text to guide readers to these details. Additionally, we plan to release our code after the double-blind review process concludes, which we hope will provide further clarity.

---

> ### Author Response · Authors · 2024-11-20
> **Thanks for your review! Authors' feedback [3/4].**
>
> **Q4 (original W4).  Code generation is only one of the many applications of agentic systems. There are plenty of other domains where LLM agents may help. It is therefore not enough to consider the task of code generation alone. The agentic evaluation framework also seems to be ad hoc to code generation.**
>
> We agree that agentic systems have applications beyond code generation. In our study, we chose code generation as a testbed for the Agent-as-a-Judge (AAAJ) framework due to its inherent complexity and the significant advancements in agentic systems in this field (e.g., having notable impact in both academia and industry recently). Developing benchmarks across multiple domains is a substantial effort, and focusing on code generation allows us to thoroughly demonstrate the effectiveness of AAAJ. For example, OpenAI's [1] from the last ICLR focuses on mathematics problems, but the idea can be applied more generally.
>
> Our intention is to demonstrate the effectiveness of the AAAJ framework in a challenging domain, with the hope that it can be extended to others in future work.
>
> [1] Lightman, Hunter, et al. (2024). "Let's verify step by step." ICLR 2024. https://openreview.net/forum?id=v8L0pN6EOi
>
> **Q5 (original Q1). Analysis of failure cases: Did you conduct any studies on the failure cases to see whether there are consistent patterns that agentic systems may exploit?**
>
> Valuable suggestion. We conducted an analysis of the failure cases and identified consistent patterns, which are summarized below:
>
> | Category                              | Count |
> |---------------------------------------|-------|
> | Data preprocessing and postprocessing | 10    |
> | Dataset or Environment                | 8     |
> | Other                                 | 5     |
> | Machine Learning Method               | 4     |
> | Performance Metrics                   | 3     |
> | Visualization                         | 3     |
> | Human-Computer Interaction            | 3     |
>
> We found that AAAJ struggled most with judging cases in the *Data preprocessing and postprocessing* category, whereas it performed well in judging *Human-Computer Interaction* cases.
>
> ---
>
> ### **Failure Case Sample 1**
>
> | Task                                          | Requirement ID | Category               | Agent-as-a-Judge | Human-as-a-Judge | Criteria                                                                                 |
> |-----------------------------------------------|----------------|------------------------|-----------------|-----------------|-----------------------------------------------------------------------------------------|
> | `40_Text_Summarization_BART_CNNDailyMail_DL` | 0              | Dataset or Environment | True            | False           | The "CNN/Daily Mail" news dataset is used, including loading and preparing the dataset in `src/data_loader.py`. |
>
> **Analysis:**
> In this case, the dataset used was a synthesized one generated by the OpenHands CodeAct agent. Human evaluators could quickly identify this discrepancy, but the agent-as-a-judge, having only checked the file path and content, was misled into believing it was the genuine CNN/DailyMail dataset.
>
> ---
>
> ###  **Failure Case Sample 2**
>
> | Task                                         | Requirement ID | Category               | Agent-as-a-Judge | Human-as-a-Judge | Criteria                                                                                       |
> |-----------------------------------------------|----------------|------------------------|-----------------|-----------------|-------------------------------------------------------------------------------------------------|
> | `46_Speech_Recognition_DeepSpeech_LibriSpeech_DL` | 2              | Machine Learning Method | True            | False           | Hyperparameters such as learning rate and batch size are tuned in `src/train.py`.              |
>
> **Analysis:**
> Here, the agent-as-a-judge confirmed that hyperparameters were set, but missed the nuance in the criteria. The requirement implied that the learning rate and batch size should dynamically adjust in `src/train.py`, something human evaluators were able to detect.

---

> ### Author Response · Authors · 2024-11-20
> **Thanks for your review! Authors' feedback [4/4].**
>
> **Q6 (original Q2). Can the AAAJ framework be evaluated in benchmarks other than code generation?**
>
> We acknowledge that our current implementation is tailored to code generation.
>
> Although we have not yet designed a general AAAJ capable of covering all domains, we plan to pursue this in the future based on your suggestion. Nonetheless, our findings indicate that AAAJ provides valuable insights both as a judge and as environmental feedback for code generation. These results suggest that AAAJ holds promising potential for generalizability for other domains by prompt and code modifications.
>
> ### Ask anything about a Python codebase (repo grokking):
>
> This setting uses a real, popular, and comparatively complex codebase for Agent-as-a-Judge to handle any queries about it.
>
> **Question**: What do you think about this project, and how can it be improved with real swarm intelligence? Please provide at least 5 suggestions with specific code examples where I need to add functions. Note that you do not need to consider the current `/examples` folder; please focus on the code in the `/swarm` directory. Thank you!
>
> **Project**: https://github.com/openai/swarm
>
> **Response:**
>
> The project appears to be structured around a core module named `swarm`, which likely serves as the main engine for the application. The `swarm` directory contains several key files: `util.py`, `core.py`, `types.py`, and `repl/repl.py`. To enhance this project with real swarm intelligence, here are five suggestions with specific code examples:
>
> 1. **Implement a Swarm Intelligence Algorithm**
>    - **File to Modify:** `swarm/core.py`
>    - **Suggestion:** Introduce a swarm intelligence algorithm such as Particle Swarm Optimization (PSO) or Ant Colony Optimization (ACO) to improve decision-making processes.
>    - **Code Example:**
>      ```python
>      def particle_swarm_optimization():
>          # Initialize particles
>          particles = initialize_particles()
>          for iteration in range(max_iterations):
>              for particle in particles:
>                  # Update particle velocity and position
>                  update_velocity(particle)
>                  update_position(particle)
>                  # Evaluate fitness
>                  fitness = evaluate_fitness(particle)
>                  # Update personal and global bests
>                  update_bests(particle, fitness)
>          return get_global_best(particles)
>      ```
>
> 2. **Enhance Communication Between Agents**
>    - **File to Modify:** `swarm/types.py`
>    - **Suggestion:** Define a communication protocol or message-passing interface to allow agents to share information effectively.
>    - **Code Example:**
>      ```python
>      class Message:
>          def __init__(self, sender_id, receiver_id, content):
>              self.sender_id = sender_id
>              self.receiver_id = receiver_id
>              self.content = content
>
>      def send_message(sender, receiver, content):
>          message = Message(sender.id, receiver.id, content)
>          receiver.receive_message(message)
>      ```
>
> 3. **Dynamic Task Allocation**
>    - **File to Modify:** `swarm/util.py`
>    - **Suggestion:** Implement a dynamic task allocation system where tasks are assigned based on agent capabilities and current workload.
>    - **Code Example:**
>      ```python
>      def allocate_tasks(agents, tasks):
>          for task in tasks:
>              best_agent = min(agents, key=lambda agent: agent.current_load)
>              best_agent.assign_task(task)
>      ```
>
> 4. **Agent Learning and Adaptation**
>    - **File to Modify:** `swarm/core.py`
>    - **Suggestion:** Introduce a learning mechanism for agents to adapt based on past experiences, using techniques like reinforcement learning.
>    - **Code Example:**
>      ```python
>      def learn_from_experience(agent, experience):
>          # Update agent's knowledge base
>          agent.knowledge_base.update(experience)
>          # Adjust strategies based on learning
>          agent.adjust_strategy()
>      ```
>
> 5. **Visualization of Swarm Activity**
>    - **File to Modify:** `swarm/repl/repl.py`
>    - **Suggestion:** Add a visualization tool to monitor and analyze the swarm's activity and performance in real-time.
>    - **Code Example:**
>      ```python
>      def visualize_swarm_activity(swarm_data):
>          import matplotlib.pyplot as plt
>          # Plot swarm data
>          plt.plot(swarm_data['time'], swarm_data['performance'])
>          plt.xlabel('Time')
>          plt.ylabel('Performance')
>          plt.title('Swarm Activity Over Time')
>          plt.show()
>      ```
>
>
> ---
>
> **Once again, we sincerely appreciate your valuable time. Your insights have been instrumental in helping us improve our work. We are committed to refining our submission based on your recommendations and look forward to any further discussions.**

---

> > ### Comment · Reviewer_qJsZ · 2024-12-03
> > **Re**
> >
> > Thank the author for studying the failure cases. My concerns on the overall writing and the limited evaluation remain after the rebuttal. So the score will be kept unchanged.

---

### Author Response · Authors · 2024-11-28

Dear all reviewers, ACs, and PCs,

According to your suggestions, we've uploaded our revised version of the paper to incorporate the comments made during the rebuttal and new experimental results we've obtained in addressing particular comments. We've carefully highlighted all the substantial changes made for this version in blue. For our camera-ready version, we will correct the color of the text.

To address reviewer qJsZ's comments
- we have updated a paragraph at the end of the introduction to clarify the structure of the paper for readers (see L107-L134);
- we have conducted a statistical analysis of failure cases (see Appendix N);
- we have provided examples for further uses of Agent-as-a-Judge, and
- we conducted experiments using different LLMs (see Appendix O).

To address reviewer MHgQ's comments
- we have provided examples for further uses of Agent-as-a-Judge, and
- we conducted experiments using different LLMs (see Appendix O).

To address reviewer 4rku's comments
- we have conducted additional Human-as-a-Judge experiments to confirm the reliability,
- we have done a word frequency analysis,
- we have resolved citation issues and corrected the bibliography, and
- we have clarified the description of the retrieval module.

We have also now added a table of contents to our appendix, as the above has brought it to a total of 30 pages.

We would like to thank reviewers 4rku and MHgQ for interacting with us during the rebuttal stage and finally recommending the paper's acceptance. We would like to thank reviewer qJsZ for their original comments that we've used to strengthen the paper. We have endeavored to thoroughly address all of your comments (e.g., clarifying the structure of the paper and adding an analysis of the failure cases in Appendix N) and would kindly ask if we've been able to successfully allay your major concerns. If so, we would kindly ask you to update your score to reflect these changes.

We again want to express our deep gratitude to all the reviewers, the ACs, and the PCs for their work so far.

---

### Meta-Review · Area_Chair_SYbo · 2024-12-19

**Metareview:**

The paper "Agent-as-a-Judge: Evaluating Agents with Agents" introduces a novel framework, Agent-as-a-Judge (AAAJ), for evaluating agentic systems (AI systems capable of autonomous decision-making and action) by leveraging other agentic systems as evaluators. This approach extends the existing LLM-as-a-Judge paradigm by incorporating intermediate feedback mechanisms, enabling more granular and accurate evaluations. To validate this framework, the authors propose a new benchmark, DevAI, consisting of 55 realistic AI code generation tasks with detailed annotations and hierarchical requirements. The paper benchmarks three leading code-generating agentic systems (MetaGPT, GPT-Pilot, and OpenHands) using AAAJ, demonstrating its alignment with human evaluators and its superior performance compared to LLM-as-a-Judge. The authors also provide supplementary materials, including failure case analyses and experiments with different backbone models, to support their claims.


Weaknesses
- Limited Scope of Benchmark : While DevAI is a valuable benchmark, its scale (55 tasks) is relatively small compared to some existing benchmarks (e.g., MLE-Bench with 75 tasks). The authors argue that increasing the number of tasks would disproportionately increase costs, but this may limit the benchmark's generalizability to a broader range of AI development scenarios.
- Task-Specific Design : The AAAJ framework is currently tailored to code generation tasks, raising questions about its applicability to other domains. Although the authors suggest its potential for generalization, this remains a theoretical claim without concrete demonstrations across diverse tasks.
- Human Evaluation Limitations : The reliance on a small pool of human evaluators (the authors themselves) raises concerns about bias and the robustness of the ground truth labels. While the authors conducted a supplementary study with a larger group of students, the scope of this additional evaluation was limited (7 tasks), leaving room for skepticism about the reliability of the human baseline.

Overall, the paper presents a compelling contribution to the field of agentic system evaluation, with the Agent-as-a-Judge framework and the DevAI benchmark offering practical and innovative solutions. While the paper has addressed many reviewer concerns through additional experiments and clarifications, some limitations persist, particularly regarding the scope of the benchmark and the framework's generalizability. The paper would benefit from further refinement in terms of presentation and broader demonstration of the framework's applicability.

**Additional Comments On Reviewer Discussion:**

Key Points and Responses During Rebuttal Period:

1. **Structural Clarity (qJsZ):**
   - **Concern**: The paper's structure was confusing, with the DevAI benchmark overshadowing the Agent-as-a-Judge (AAAJ) framework.
   - **Response**: The authors added a clarifying paragraph in the introduction to outline the logical flow from benchmark creation to evaluation challenges and the AAAJ solution.

2. **Separation of Contributions (qJsZ):**
   - **Concern**: The AAAJ framework and DevAI benchmark should be separated into distinct papers for more detailed discussions.
   - **Response**: The authors argued that the contributions are intrinsically linked, providing a rationale for their combined presentation and suggesting potential future directions.

3. **Human Evaluation Methodology (4rku):**
   - **Concern**: The use of only three human evaluators (the authors themselves) raised concerns about bias and insufficient robustness.
   - **Response**: The authors conducted an additional study with 10 external evaluators on a subset of tasks, demonstrating alignment with the original results and reinforcing the reliability of their baseline.

4. **Inconsistent Statements on Reliability (4rku):**
   - **Concern**: The paper contained conflicting statements about the reliability of human evaluations.
   - **Response**: The authors clarified that human evaluations were treated as a reliable baseline despite inherent limitations and explicitly outlined the relative reliability of evaluation methods.

5. **Benchmark Scale and Complexity (MHgQ):**
   - **Concern**: The DevAI benchmark was criticized for its small scale (55 tasks) and potential oversimplification of tasks.
   - **Response**: The authors justified the benchmark size by comparing it to concurrent benchmarks and emphasizing its high-quality design, evidenced by low completion rates by tested agents.

6. **Technical Novelty of AAAJ (MHgQ):**
   - **Concern**: AAAJ was viewed as a natural extension of LLM-as-a-Judge, lacking significant technical novelty.
   - **Response**: The authors outlined specific technical innovations, such as automated evidence collection and modular design, and provided a detailed example of AAAJ's evaluation process.


Despite the authors' efforts to address reviewer concerns, the paper still exhibits some limitations that outweigh its contributions. The structural issues, while partially addressed, continue to hinder the readability and coherence of the manuscript. The limited scope of the DevAI benchmark and the lack of demonstrated generalizability of the AAAJ framework beyond code generation tasks remain substantial weaknesses.

---

### Decision · Program_Chairs · 2025-01-22

Reject